# Synovial Sarcoma Preclinical Modeling: Integrating Transgenic Mouse Models and Patient-Derived Models for Translational Research

**DOI:** 10.3390/cancers15030588

**Published:** 2023-01-18

**Authors:** Lorena Landuzzi, Francesca Ruzzi, Pier-Luigi Lollini, Katia Scotlandi

**Affiliations:** 1Experimental Oncology Laboratory, IRCCS Istituto Ortopedico Rizzoli, 40136 Bologna, Italy; 2Laboratory of Immunology and Biology of Metastasis, Department of Medical and Surgical Sciences (DIMEC), University of Bologna, 40126 Bologna, Italy

**Keywords:** synovial sarcoma, conditional mouse models, patient-derived xenografts (PDX), epigenetic drugs, immunotherapy

## Abstract

**Simple Summary:**

Synovial sarcoma (SyS) is a rare malignant tumor mainly occurring in children, adolescents, and young adults. SyS displays the pathognomonic t(X;18) translocation resulting in the SS18-SSX fusion protein being able to interact with both the BAF enhancer complexes and polycomb repressor complexes, and either activate or repress gene transcription, resulting in genome-wide epigenetic deregulation and altered gene expression. This review analyzes the different experimental in vivo models for SyS research: (I) conditional transgenic mouse models expressing the SS18-SSX fusion protein that, alone or combined with some of the few other recurrent alterations (gains in BCL2, Wnt-β-catenin signaling, FGFR family, or loss of PTEN and SMARCB1), spontaneously develop SyS; (II) SyS patient-derived xenografts (PDX) established in immunodeficient mice; (III) SyS cell lines and cell line-derived xenografts. SyS preclinical models are greatly contributing to the disclosure of additional vulnerabilities and to the development of new therapeutic approaches for SyS.

**Abstract:**

Synovial sarcomas (SyS) are rare malignant tumors predominantly affecting children, adolescents, and young adults. The genetic hallmark of SyS is the t(X;18) translocation encoding the SS18-SSX fusion gene. The fusion protein interacts with both the BAF enhancer and polycomb repressor complexes, and either activates or represses target gene transcription, resulting in genome-wide epigenetic perturbations and altered gene expression. Several experimental in in vivo models, including conditional transgenic mouse models expressing the SS18-SSX fusion protein and spontaneously developing SyS, are available. In addition, patient-derived xenografts have been estab-lished in immunodeficient mice, faithfully reproducing the complex clinical heterogeneity. This review focuses on the main molecular features of SyS and the related preclinical in vivo and in vitro models. We will analyze the different conditional SyS mouse models that, after combination with some of the few other recurrent alterations, such as gains in BCL2, Wnt-β-catenin signaling, FGFR family, or loss of PTEN and SMARCB1, have provided additional insight into the mechanisms of synovial sarcomagenesis. The recent advancements in the understanding of SyS biology and improvements in preclinical modeling pave the way to the development of new epigenetic drugs and immunotherapeutic approaches conducive to new treatment options.

## 1. Introduction

Synovial sarcomas (SyS) are rare malignant tumors accounting for 5–10% of all soft tissue sarcomas (STS) [1], occurring mainly in children, adolescents, and young adults (AYA, age 15–39). SyS is the second most common STS in the pediatric population after rhabdomyosarcoma, and represents the second most frequent STS in AYA after mixoid/round liposarcoma [2]. Clinical presentation in the soft tissue of the lower limb (45–75%) [3,4,5], around the knee and the ankle, is the most frequent, followed by the upper limb and the thorax. Less common sites are the head and neck region, abdominal wall and retroperitoneum, pelvis and trunk, lungs, and heart. Despite some histological resemblance to synovial tissue, SyS does not directly arise from synoviocytes. The widespread distribution of SyS and the uncertain differentiation make the precise cell of origin of SyS still controversial, however, the prevalent onset in proximity of joints, bones [6], and skeletal muscles suggests a multipotent mesenchymal stem cell origin.

Around 50% of the cases are localized at the presentation [5], but late metastases occur in 50–70% of patients, mainly in the lungs (80%), lymph nodes, and bone marrow, with occasional involvement of bone and liver [7].

Standard of care in primary localized SyS consists of wide surgical resection combined with radiotherapy when appropriate [8]. However, radiosensitivity of SyS is low and the response rate does not exceed 50% [9]. For high-risk patients, cytotoxic chemotherapy, both as neoadjuvant and adjuvant treatment, is based on anthracyclines and ifosfamide and, in second line, on trabectedin and pazopanib able to inhibit VEGFR 1, 2, and 3; PDGFRa and b; c-Kit; and, at a lower extent, FGFR 1 and 3 [7,8,10]. Overall survival (OS) rate for localized disease at diagnosis is around 70–80% at 5 years and around 60% at 10 years, while metastases can occur very late, even over 10 year later [4,5,7]. In contrast, OS rate of patients with metastatic disease at diagnosis does not exceed 10–20% at 5 years [4,5,7]. The role of chemotherapy in SyS is not very well defined. Pediatric or adolescent patients with low-risk tumors (localized small low-grade tumors) can be successfully managed with surgery alone, without systemic therapy [7]. Overall, compared to other STS, SyS seems to be more chemosensitive, although data are still controversial, particularly for adult patients [8]. In general, cytotoxic drugs appear to be more effective in children, with a response rate of 50–60%, than in adults, where the response rate is around 30% [11,12,13].

From a morphological point of view, different subtypes have been identified. The most frequent (50–60% of all SyS) is the monophasic subtype, showing a majority of small tightly-packed spindle cells. A monophasic epithelioid subtype, displaying a uniform glandular pattern, is rarely reported. The biphasic subtype is the second most common histotype, representing 20–30% of SyS; it is characterized by the presence of both epithelial and spindle cells arranged in alveolar- or gland-like structures or solid nests. The poorly differentiated variant accounts for 10–20% of SyS and consists of small round cells. Large cell epithelioid variants and high-grade pleomorphic spindle cell variants are also recognized [4,7]. The most unfavorable prognosis has been ascribed to the epithelioid-type, while the monophasic spindle cell variant has an intermediate prognosis and the biphasic type show the best prognosis [4,5,7]. Of note, most SyS show cytoplasmic Bcl-2 overexpression and express both epithelial markers, such as cytokeratins, and mesenchymal markers, such as vimentin [14,15].

From a genetic point of view, the genetic hallmark of SyS is a balanced t(X;18) chromosomal translocation that gives rise to the fusion of almost the whole SS18 gene (previously termed SYT) on chromosome 18q11 to a portion of either SSX1, SSX2, or, more rarely, SSX4 genes on chromosome Xp11 [16,17,18,19,20,21]. The SS18-SSX1 fusion is usually associated with a biphasic histology, while SS18-SSX2 is more frequent in the monophasic subtype, but correlations with prognosis are controversial [8]. Neither SS18, nor SSX possess DNA binding domains, however, their interaction with transcriptional and epigenetic regulators allows SS18-SSX to elicit a wide deregulation of gene expression. The fusion protein integrates, by means of the SS18 component, into the BAF family complexes (barrier-to-autointegration factor, also known as BRG1-associated factor or mammalian SWI/SNF), multimeric protein structures existing in three final-form complexes (canonical BAF, CBAF; polybromo-BAF, PBAF; non-canonical BAF, ncBAF or GBAF) having crucial roles in chromatin organization. The fusion protein, by inducing imbalance in BAF family complexes, can alter chromatin remodeling and activate aberrant gene transcription [22,23]. The SSX component mediates the interaction with the polycomb chromatin repressor complexes, PCR1 and PCR2, [24,25] involved in gene transcription inhibition. The fusion oncogene, by inducing a broad transcriptional dysregulation, including, for example, Sox2 activation required for SyS proliferation, represents the major driver of transformation and malignancy. Additional genomic alterations (p53 or H-Ras mutations, MDM2 amplification, or PTEN loss) are few and infrequent [8,24]. Gene expression and DNA methylation profiling have provided a distinct SyS sarcoma signature that makes them cluster separately from other STS and from other fusion-driven tumors [23,26,27,28].

This review will describe the many in vivo and in vitro SyS models available for preclinical studies and their contribution to the understanding of SyS molecular hallmarks and dependencies that could lead to the development of new epigenetic drugs and therapeutic strategies for SyS.

## 2. Modeling SyS in the Mouse

Experimental oncology has gained a great advantage from the possibility of modeling human tumors in the mouse, contributing to the understanding of molecular mechanisms of tumorigenesis and to the development of new therapeutic approaches. Preclinical models range from syngeneic mouse models accepting the growth of mouse tumor cell lines, for example, derived from mouse melanoma, mammary, lung, or colon cancers, or sarcomas, to genetically engineered mouse models (GEMM), for example, transgenic mouse models carrying unique activated oncogenes or deleted for oncosuppressor genes and spontaneously developing genome-driven tumors.

Recently, conditional transgenic mouse models expressing oncogenic human fusion genes under the control of selected promoters, or severe immunodeficient mice permissive to the growth of in vitro-cultured human tumor cells or tumor fragments directly obtained from patients (patient-derived xenografts, PDX) [29,30], have greatly expanded the gamut of preclinical models available for translational research.

At variance with other human tumors that lack of a matching mouse model, in recent years, SyS research can benefit from several of these different models. Scientists have been able to obtain, in immunocompetent mice, conditional transgenic mouse models spontaneously developing SyS closely resembling human SyS, and, in immunodeficient mice, the growth of stable SyS PDXs maintaining the molecular and histological features of the patient tumor of origin and preserving the intra- and inter-tumor heterogeneity observed at clinical level. Despite some limitations, regarding, for example, difficulties in reproducing in the conditional mouse models the histopathological and clinical behavior of human SyS and the exact natural history of the disease, due to the absence of knowledge of the cell of origin, or, in the immunodeficient mice, difficulties related to the absence of the immune response, the integration of data from these different research models deserves careful consideration in their potentiality of improving the understanding and the therapeutic approaches for SyS.

## 3. SyS Conditional Transgenic Mouse Models

Many attempts have been made to develop mouse models of other pediatric solid tumors carrying fusion genes, but most of them were unsuccessful or did not achieve the expected tumor occurrence. SyS is one of the few human tumors induced by a unique specific gene translocation that researchers have been able to reproduce in transgenic mice [31,32,33,34,35,36] with high tumor histotype fidelity and obtaining high tumor penetrance.

In 2007, Haldar and coworkers created and maintained, on a mixed C57BL/6 and 129/SvJ background, a mouse model for SyS by conditional expression of the fusion gene SS18-SSX2 (termed hSS2 or MSS2) [37]. They designed a conditional SS18-SSX2 containing vector for ROSA26 targeted transgenesis, which was inserted into the mouse chromosome 6 ROSA26 locus, ubiquitously active. Between the ROSA26 promoter and SS18-SSX2, a strong transcriptional termination signal, NeoPA, flanked by loxP sites (floxed, fl), was inserted. Cre recombinase deletes DNA that is flanked by loxP sites, and removal of the STOP cassette by Cre allows for expression of the fusion gene. This means that, in the absence of Cre protein, SS18-SSX2 in not transcribed; when Cre protein is produced, determining Cre-dependent recombination of loxP sites and NeoPA removal, the SS18-SSX2 fusion transcript is expressed.

In a different mouse strain, Myf5–Cre mice, Cre protein production was induced in committed myoblasts expressing the myogenic regulatory factor Myf5 by introduction of a Myf5-targeting vector fused to the Cre recombinase cDNA. Mice of each individual transgenic strain, both homozygous and heterozygous, were normal, viable, and fertile, with no tumor onset for more than one year.

When the conditional SS18-SSX2 mice were bred with Myf5–Cre mice, about 8% of the progeny died within 2 months with smaller body dimensions but no tumors, and the rest developed multiple synovial sarcomas by the age of 3–5 months with complete penetrance (100%).

Synovial sarcomagenesis was obtained also by using a tamoxifen-inducible CreER system. CreER is a fusion between Cre and a mutant estrogen receptor that keeps Cre protein in the cytoplasm; exogenous treatment with tamoxifen allows nuclear translocation of Cre and recombination activity [38]. After tamoxifen injection, in CreER/hSS2 mice, synovial sarcomagenesis showed complete penetrance with 5–14 months tumor latency and high tumor multiplicity. SyS formation was observed even in the absence of tamoxifen administration, due to the leakage of the system. SyS distribution, although always near to skeletal structures, was slightly different from that observed in hSS2/Myf5–Cre mice, intriguingly suggesting a non-myogenic origin for synovial sarcomas. Mouse SyS and human SyS shared both histological features and molecular transcription profiles that led to the identification of a set of 72 genes as a ‘‘synovial sarcoma signature’’ [37,38].

Remarkably, early and ubiquitous expression of the fusion protein, obtained by cross-breeding with *hypoxanthine guanine phosphoribosyl transferase* (*Hprt)–Cre* mice, caused embryonic lethality; similarly, expression of the fusion protein in early precursors in the myogenic lineage, PAX3- and PAX7-expressing progenitors, achieved by cross-breeding with *Pax3–Cre* and *Pax7–Cre* mice, disrupted normal embryogenesis. Interestingly, expression of the fusion protein in more differentiated myocytes and myofibers, obtained by crossbreeding with *Myf6*–Cre mice, did not induce tumors, but gave rise to severe myopathy [37]. Overall, genetically modified mice demonstrated that most cell lineages cannot tolerate the expression of the fusion oncogene and are not able to survive after its expression; only intracellular and/or microenvironmental permissive conditions can lead to cell survival and neoplastic transformation.

The Myf5–Cre/hSS2 conditional mouse model appeared to indicate that, in the skeletal muscle lineage, immature but committed myoblasts or quiescent satellite cells can provide a permissive environment for transformation by the fusion oncogene SS18-SSX2 and the onset of SyS [37], while earlier precursors or more differentiated myogenic elements do not. Myf5–Cre/SS18-SSX2 elements were viable near cartilages, and the microenvironment near cartilages appeared to prevent apoptosis of Myf5–Cre/SS18-SSX2 cells, therefore predicting tumor onset near joints [37]. A further analysis of the embryonic expression of Myf5–Cre suggested that the lineage includes also some osteochondroprogenitor cells that could be the target of the transforming potential of the fusion oncogene. The effect of the fusion oncogene in early osteoblast precursors was explored by conditional activation of the hSS2 allele at various stages of osteoblast differentiation with Cre protein expression controlled by different osteogenic differentiation drivers. The activation of late preosteoblast-osteoblast differentiation-specific promoters, such as Col1a1Cre or OcCre, resulted in almost complete perinatal lethality, but no tumor formation [6]. To elude embryonic or developmental toxicity of SS18-SSX expression across all lineages, the tamoxifen-inducible Cre recombinase system was used. After administering tamoxifen to OsxCreERT hSS2 mice, no sarcoma formation was observed for over one year. On the contrary, tamoxifen-treated Prx1Cre^ERT2^hSS2 mice, under the control of a promoter active in early mesenchymal precursors, developed perimandibular SyS by 9 months of age, indicating that SS18-SSX expression was allowed by differentiating osteoblast precursors and was able to transform the more primitive elements [6].

The prevalent SyS translocation, SS18-SSX1 (termed hSS1), was also able to induce SyS in mice. Conditional hSS1 mice [39] were bred with the Myf5–Cre mice, and the Myf5–Cre/hSS1 progeny displayed almost complete penetrance of synovial sarcomagenesis. Compared to hSS2, the hSS1 allele was slightly less sarcomagenic, showing a longer median tumor latency (over 1 year) and a lower tumor multiplicity. In both Myf5–Cre/hSS1 and Myf5–Cre/hSS2 lineages, tumor onset, equal between genders, was mainly within skeletal muscle due to Myf5-specific expression in skeletal muscle-committed cells. Tumor histology resembled human synovial sarcomas with most monophasic subtypes and a lower proportion of biphasic morphologies, with no significant differences between the two different translocations.

Synovial sarcomagenesis was obtained also after TATCre protein injection in the hind limb, but with incomplete penetrance for both alleles and over 1 year tumor latency. Transcriptomes of tumors of the two fusion types differed only minimally. Exome sequencing did not identify additional relevant somatic mutations, except for frequent amplification of the fusion expression locus on chromosome 6, suggesting a dose-dependent effect on tumorigenesis. In agreement with these data, after TATCre protein injection in the hind limb at 4 weeks of age, homozygous *hSS2* mice demonstrated higher (complete) penetrance of tumorigenesis and a faster tumor growth compared to heterozygous mice [23]. The available conditional SyS mouse models, their main features, and their role in the understanding of synovial sarcomagenesis are summarized in Table 1.

Silencing or activation of the most relevant oncogene and tumor suppressor gene reported to be altered in SyS has helped in defining their role in SyS onset and in the disclosure of further oncogenic driver mechanisms and dependencies. This meticulous work has led even to the development of a spontaneous SyS metastatic model as described in the following paragraphs.

### 3.1. Role of Proto-Oncogenes Explored in Conditional SS18-SSX Transgenic Mouse Models

#### 3.1.1. Bcl-2 Family

BCL2 is an antiapoptotic oncogene highly expressed in human SyS. The Bcl-2 family is characterized by the presence of one or more BCL2 homology (BH) domains and includes pro-survival (BCL2, BCL-XL, and MCL-1), BH3-only pro-death (BIM, NOXA, PUMA), and terminal-effector (BAK and BAX) molecules. The pro-survival proteins bind and sequester the BH3-only activator proteins and prevent the activation of the effectors BAK and BAX and the initiation of mitochondrial pore formation and apoptosis. Other members of the Bcl-2 family, apoptotic and antiapoptotic genes, are also dysregulated in SyS, but at lower levels compared to BCL2 [40]. When conditional hSS2 mice were crossed with mice overexpressing Bcl-2 (Bcl-2OE mice), and TATCre protein injection in the hind limb was used to induce expression of both SS18-SSX2 and BCL2 genes, Bcl-2 overexpression significantly enhanced synovial sarcomagenesis, and the same effect was observed after crossbreeding with Myf5–Cre/hSS2 mice [41]. On the contrary, BCL2 conditional deletion in Myf5–Cre/hSS2, Bcl2fl/fl mice mildly disrupted synovial sarcomagenesis, inducing a slightly later onset and lower multiplicity. The development of tumors showing absence of Bcl-2 protein indicated that Bcl-2 is not strictly required for SS18-SSX2-mediated sarcomagenesis. In vivo, against mouse SyS, specific pharmacological inhibition of Bcl-2 by ABT-199 (venetoclax) treatment was ineffective. Overall, Bcl-2 seems to contribute to synovial sarcomagenesis, but it does not represent an absolute dependency and its targeting is trickier. Conversely, targeting BCL-XL, a mitochondrial antiapoptotic Bcl2 family member, with the specific BCL-XL inhibitor BXI-72, achieved a significant inhibition of tumor growth in the conditional SyS mouse model, indicating this molecule as a possible target in SyS [41]. Similarly, the use of ABT-263, a BH3-peptidomimetic inhibitor of BCL2, BCL-XL, and BCLW, despite a limited activity in vitro as a single agent against mouse SyS cells, was able to hamper synovial sarcomagenesis in vivo in the SyS mouse model Myf5–Cre/hSS2, significantly reducing tumor number and dimensions [42]. Finally, in a recent study, since human SyS tumors, compared to other translocated sarcomas, demonstrate increased expression of BCL2 and decreased expression of NOXA, the endogenous inhibitor of MCL-1, the resulting increased activity of MCL-1 pro-survival protein was identified as responsible for venetoclax resistance in SyS. Only co-targeting of both BCL-2 and MCL-1 proved to be effective against both a human SyS cell line-derived and an SyS PDX cell line-derived animal model, possibly supporting the clinical evaluation of the combined approach [40].

#### 3.1.2. WNT/β-Catenin Signaling

Wnt/β-catenin signaling controls embryogenesis and is aberrantly activated in tumorigenesis. β-catenin (CTNNB1) is a proto-oncogene with a dual role being involved in cell adhesion and, after nuclear localization, in gene transcription. Nuclear β-catenin staining has been reported in 30–70% of SyS, and deregulation of Wnt signaling, whether due to APC-inactivating and CTNNB1-stabilizing mutations that are rare (around 8% each) or epigenetic mechanisms, have been detected in the majority of human SyS. Therefore, the role of β-catenin in SyS was explored by the conditional silencing of the proto-oncogene β-catenin (B-CAT^fl^) in hSS2/Myf5–Cre mice. The inhibition of Wnt signaling through the genetic loss of β-catenin was able to block SyS formation: homozygous hSS2+/B-CATfl+/+/Myf5–Cre+ mice showed a dramatic inhibition of synovial sarcomagenesis compared to hSS2/Myf5–Cre mice [43]. On the contrary, a significant increase in SyS incidence was obtained in hSS2 mice after induction of β-catenin nuclear stabilization [44]. The Axin–APC–GSK3β complex modulates β-catenin activity through phosphorylation, targeting it for ubiquitination and proteasomal degradation. Prevention of its phosphorylation by canonical Wnt signal transduction, APC inactivating mutations, or mutations in the third exon of β-catenin stabilize β-catenin to the nucleus. In Rosa26hSS2/wt; Ctnnb1ex3fl/wt mice heterozygous for a beta1 catenin allele having a floxed exon3 sequence, Cre-mediated recombination generates *Ctnnb1Δex3*, which expresses a stabilized β-catenin with conserved trans-activation activities and increased nuclear localization. In these mice, adenovirus–Cre (AdCre) vectors or TATCre protein administration in limb muscle adjacent to bone greatly enhanced synovial sarcomagenesis, with tumors developing within 3 months after Cre treatment. Tumors were mainly poorly differentiated SyS, thus suggesting that β-catenin stabilization could hamper epithelial differentiation.

In human SyS samples, activating genetic mutations of Wnt pathway components are rare, suggesting that constitutive activation of Wnt/β-catenin in synovial sarcomas is caused by upstream mechanisms. Domain deletion analysis provided evidence that SS18-SSX2 activates Wnt/β-catenin signaling through the domains that mediate its function as a coregulator of gene expression, both through the SSXRD domain that mediates SS18-SSX interaction with polycomb components and disrupts their gene silencing function, and through the first 40 amino acids of the SS18 amino (N)-terminal region, which associates with BAF complexes, activators of gene transcription [43].

Deregulation of β-catenin appears to be a constant trait of SS18-SSX2-induced SyS tumors and is necessary for their development, highlighting a strong Wnt/β-catenin signaling dependency in SyS. The use of Wnt inhibitors in vivo was able to hamper SyS growth in conditional mouse SyS models and in xenograft SyS models, indicating the Wnt/β-catenin pathway as a possible target for SyS therapy [43].

Of note, in the mutated *β-catenin* mouse model, TatCre injection in the subcutaneous tissue of the abdominal wall did not induce any tumor. Only TatCre injection in proximity of bone induced *β-catenin*-stabilized SyS development. Compared to Myf5–Cre/hSS2 SyS, β-catenin-stabilized SyS expressed higher levels of osteoprotegerin (OPG), a secreted osteoclastogenesis inhibitory factor, which is abundant in bones. The role of OPG in synovial sarcomagenesis was further explored by exogenous administration of OPG, which enhanced SyS growth, and in hSS2 mice with a genetic loss of function of OPG (Opg−/−), in which tumors, after TATCre injection, developed similarly to the OPG wild type mice, but showed a reduced growth rate. In these experiments, OPG appeared to be not necessary for synovial sarcomagenesis, but able to enhance SyS growth, thus providing a paracrine link between the bone and synovial sarcomagenesis [6].

In addition, the Ctnnb1ex3fl allele was combined also with CreERT under the control of promoters active at various stages of osteoblast differentiation. In Prx1CreERT2 hSS2 Ctnnb1ex3fl mice, tamoxifen administration led to dramatic synovial sarcomagenesis deriving from the periosteal surface of the forelimb bones. Even OsxCreERT hSS2 (previously shown not to be able to develop SyS), after the combination with the Ctnnb1ex3fl gene in the triple-heterozygous mice, formed SyS deriving from the bone surface with 100% incidence. Overall, conditional mouse models indicated that OPG and β-catenin stabilization could support synovial sarcomagenesis from preosteoblasts, thus providing additional insight into the cell of origin of SyS related to bone and mesenchymal progenitors and explaining the age distribution of SyS, since OPG secretion and mesenchymal progenitor cells are at the highest levels in children, adolescents, and young adults [6].

#### 3.1.3. Fibroblast Growth Factor Receptor (FGFR) Family and ETV4/ETV5 Transcription Factor

Deregulated expression of fibroblast growth factors (FGFs) and their receptors sustaining FGF autocrine loops has been found in human SyS samples and cell lines, suggesting a relevant role in SyS growth, both in vitro and in vivo [45,46]. Consequently, the role of FGFR signaling in the pathogenesis of SyS was studied in FGFR-knockout models by creating triple-mutant hSS2/Myf5–Cre Fgfr(1,2,3)^fl/fl^ mice [47]. Homozygous deletion of FGFR1,2,3 alleles significantly inhibited SyS formation, with FGFR2 playing a major role in the mouse model.

Pharmacological inhibition of FGFRs by means of the selective inhibitor of FGFR1, FGFR2, and FGFR3, BGJ398, was able to hamper SyS growth in vivo in a model of human SyS cell-derived xenografts (SYO-1 cells) and in the conditional SyS mouse model hSS2/Myf5–Cre, where a marked decrease in average tumor number and volume was observed [47].

Interestingly, in the same study, BGJ398 treatment in SYO-1 and HS-SY-II cells was shown to abrogate the MAPK-ERK section of the FGFR pathway by a near-complete depletion of phosphorylated ERK1/2, without affecting the PI3K pathway, since the inhibitor failed to alter phospho-AKT levels. Gene and protein expression analysis of BGJ398-treated and untreated human SyS cells showed a marked downregulation of four FGFR signal-related genes: SPRY4 and DUSP6 negative regulators of FGFR/MAPK-ERK signaling, and ETV4 and ETV5, belonging to the PEA3 subfamily of oncogenic ETS transcription factors. Then, ETV4 and/or ETV5 were found to be overexpressed both in human SyS samples and in mouse hSS2/Myf5–Cre SyS samples and appeared to be differentially regulated by specific members of the FGFR family. Exogenous expression of the fusion oncogene SS18-SSX was able to upregulate various FGF ligands/receptors and ETV4 and ETV5 expression that act as downstream targets. Conversely, silencing of either ETV4 or ETV5 in SYO-1 cells led to significant inhibition of tumorigenicity in vivo, underlining a key role of both molecules in synovial sarcomagenesis. Upon ETV4 and ETV5 knockdown, upregulation of the embryonic DUX4 pathway leading to cell death was observed. These data supported the hypothesis that autocrine FGF/FGFR signaling initiated by SS18-SSX activates ETV4 and ETV5, which trigger E2F and promote cell cycle progression and expression of CHAF1A/B (chromatin assembly factor 1A/B, a major chromatin repressive complex that keeps the DUX4 gene in a silenced state of expression), leading to suppression of the DUX4 program. ETV4 and ETV5 depletion results in reversal of this oncogenic axis, DUX4 activation, and cell death. Overall, the inhibition of all three FGFRs, the targeting of ETV4 and ETV5, and the activation of the DUX4 atrophy program that, in SyS, seems to have tumor suppressor functions that appear to represent promising therapeutic approaches for SyS [47].

The main conditional SyS mouse models carrying altered oncogene expression are summarized in Table 2.

### 3.2. Role of Tumor Suppressor Genes in Conditional SS18-SSX Transgenic Mouse Models

#### 3.2.1. PTEN and the Phosphatidyl Inositol (PI)3′-Lipid Pathway

A metastatic mouse SyS model was obtained by silencing the oncosuppressor gene PTEN (phosphatase and tensin homologue on chromosome 10) that inhibits the phosphatidyl inositol(PI)3′-lipid pathway [48]. In human SyS, epigenetic downregulation of PTEN [49,50] or loss of PTEN [26] have been reported as secondary genetic changes related to tumor progression and metastatic spread. Indeed, deletion of PTEN, by means of crossbreeding with mice carrying a floxed PTEN allele, in conditional mouse models of locally-induced expression of SS18-SSX1 or SS18-SSX2 after TATCre injection, strongly increased synovial sarcomagenesis. In mouse SyS, PTEN silencing induced an inflammatory transcriptome and increased vascularization and myeloid-derived cell infiltration. The enhanced angiogenesis and inflammation promoted spontaneous lung metastasis formation, with 40–70% of the mice displaying respectively overt lung metastasis or micrometastases [48]. The mouse SyS model, offering a fully competent immune system, enabled the identification of the recruitment of monocyte/macrophages and neutrophils in the tumor microenvironment in response to the production of CSF1 by tumor cells through enhanced PI3′-lipid/pAKT signaling as the mechanism responsible for increased tumor growth and metastatic dissemination. Treatment with a selective inhibitor of CSF1R, BLZ945, inhibited tumor growth and metastases, thus suggesting that targeting inflammatory cells can impact SyS growth and malignancy.

Of note, the complete tumor penetrance and the fast SyS tumor growth of hSS2/PTEN^fllfl^ mice after TATCre injection in the hind limb made this strain useful also for in vivo drug testing, for example, demonstrating the anti-tumor activity of the HDAC inhibitor quisinostat [51,52].

#### 3.2.2. SMARCB1 and BAF-Family Complex Dysregulation

The observation that SyS frequently displays low or absent immunohistochemical staining for SMARCB1 (also known as BAF47, INI1, SNF5) led to investigations on the role of SMARCB1 in synovial sarcomagenesis by creating new conditional transgenic mouse models combining SMARCB1 genetic loss with SS18-SSX expression. A mouse strain carrying a SMARCB1 floxed allele (SMARCB1^fl^) that is excised in the presence of Cre recombinase was created [23]. In these mice, genetic silencing of SMARCB1 alone in the mesenchyme could induce tumors with incomplete penetrance and very long latency. When SMARCB1 fl mice were crossbred with hSS2 mice expressing the SS18-SSX fusion gene, sarcomagenesis was significantly enhanced with 100% penetrance and shorter tumor latency, however, tumor histology and molecular profile were more similar to epithelioid sarcomas or malignant rhabdoid tumors (MRTs), two tumor histotypes both characterized in patients by homozygous loss of function of SMARCB1. By genetic analysis, both human SyS and mouse SyS expressing SS18-SSX clustered very closely and were distinct from epithelioid and MRT tumors or SMARCB1-silenced tumors [23], thus indicating that the presence of SMARCB1 contributes to the transcriptional and phenotypic characteristics of SyS.

In the same study, a deep analysis of SS18-SSX and SMARCB1 protein fate in BAF-family complex assembly, integrating data from mouse and human SyS and SMARCB1-deleted tumors, led to a more comprehensive proposal about the mechanisms of BAF-complex dysregulation in SyS. In vivo, in SyS, incorporation of SS18-SSX into CBAF complexes leads to whole-complex CBAF degradation. The observed reduced levels of SMARCB1 in SyS derive from the whole-complex degradation of CBAF with relative increases in the abundance and prevalence of other BAF-family subtypes, PBAF and GBAF complexes. Thus, SS18-SSX alters BAF subtype levels/balance and their genome distribution, driving in this way synovial sarcomagenesis.

SS18-SSX-induced CBAF complex degradation is supposed to have a broad impact on the cell that, if able to survive CBAF reduction, upregulates GBAF and PBAF. GBAF seems to be the prevalent SS18-SSX-containing BAF-family complex on SyS chromatin. Drugs targeting GBAF and leading to its degradation, not simply inhibition, by interacting with the specific GBAF component BRD9 (bromodomain-containing protein 9), have been able to significantly hamper SyS growth in vitro and in vivo in preclinical studies [53], and two new selective protein degraders of BRD9, FHD-609 and CFT8634, have recently entered clinical development in phase I clinical trials (ClinicalTrials.gov Identifier: NCT04965753 and NCT05355753, respectively) [54]. Of note, as suggested by Li and coworkers [23], mechanisms of resistance to these new therapeutic strategies will most likely involve downregulation of BAF-complex proteasomal degradation.

Table 3 summarizes the main conditional SyS mouse models carrying altered tumor suppressor gene expression and the main studies using these models.

### 3.3. Future Developments and Applications of GEMM in SyS Research

Genetically engineered mouse models based on site-specific recombinase technology are fundamental tools for the understanding of the pathogenesis and molecular biology of cancer, as in the case of SyS, but can be expensive and time-consuming. Huang and coworkers have demonstrated that CRISPR-Cas9 technology can be used to generate multiple subtypes of soft tissue sarcomas (undifferentiated pleomorphic sarcoma and malignant peripheral nerve sheath tumor) in mice. Primary sarcomas generated with CRISPR-Cas9 and Cre recombinase technology had similar histology, growth kinetics, copy number variation, and mutational load [55], and it is possible that, in the future, also SyS in vivo modeling will benefit from the CRISPR-Cas9 technology.

Overall, the immunocompetent conditional mouse models of SyS have provided a valuable contribution to the comprehension of oncogenic mechanisms of the disease and can represent trustable preclinical tools for the development of new therapeutic approaches including epigenetic and immunological strategies.

It is frequently observed that many drugs demonstrating antitumor activity against cell lines in vitro or cell line-derived xenografts in immunodeficient mice fail to be active in mice carrying PDX, or in conditional mouse models spontaneously developing tumors. For example, doxorubicin, which is active against most SyS cell lines in vitro, but demonstrates only subtle cytoreduction in a minority of human synovial sarcoma patients, had similar response rates in the SyS conditional mouse model [41]. For this reason, when appropriate, GEMM are becoming strongly recommended also among the Minimum Preclinical Testing Requirements for the Development of Innovative Therapies [56] that recommend accurate integration of data from different preclinical models: cell lines in vitro, cell line-derived xenografts, GEMM, and patient-derived xenografts.

SyS syngeneic models in immunocompetent mice could be particularly relevant for further developments in immunotherapeutic approaches. Clinical trials reported disappointing results with the use of immunological checkpoint inhibitors (pembrolizumab anti-PD-1, ipilimumab anti-CTLA4, nivolumab anti-PD1) as monotherapy in SyS [8,57,58,59,60]. However, SyS has been found to express high levels of cancer-testis antigen genes (CTAG1A encoding NY-ESO-1, PRAME, and MAGEE1), usually expressed in tumors or in the germline, but not in normal adult tissues, as well as of receptor tyrosine kinases PDGFRA, EGFR, and ERBB2 [24]. All these potentially immunogenic proteins are receiving increasing attention because they could represent the target of new immunotherapeutic approaches in SyS.

Immunotherapeutic strategies based on the transfer of specific peptide-enhanced affinity receptor (SPEAR) T cells directed against the NY-ESO-1 antigen are currently being evaluated in SyS therapy [61] and have reached promising results in clinical trials [59,61,62,63].

In this context, immunocompetent SyS mouse models could provide a valuable tool for exploring additional immunotherapeutic strategies and identify molecular determinants of sensitivity/resistance to immunological therapies able to predict the clinical outcome.

## 4. SyS Patient-Derived Xenograft (PDX) Models

Animal models allowing the expansion of human neoplasms are essential tools more for the study of rare tumors, also because of the paucity of clinical samples. PDXs represent mouse models of human tumors created by direct engraftment of fresh surgical tumor fragments into immunodeficient mice. Methods of PDX obtainment, PDX ability to preserve patient tumor phenotype/genotype/molecular profile, and to predict therapeutic response, as well as PDX advantages and disadvantages, have been extensively illustrated in many research and review articles [29,64,65,66,67,68,69]. The availability of PDX models together with databases that collect their molecular profile are essential tools for PDX-based studies, thus several international PDX repositories [69,70,71] have been created in recent years, and guidelines describing the minimal information for the standardization of PDXs (PDX-MI) [72] have been provided.

PDX models have been obtained for most relevant bone sarcomas [29,65,66] and soft tissue sarcomas, including SyS [73,74]. Several research articles report the use of an SyS PDX for drug testing [74,75,76], however, the inter-tumor heterogeneity of SyS suggests that no single model system will be effective to test the therapeutic potential of specific drugs across the SyS variegated clinical landscape. Reliable SyS PDXs up to now are not as many as those offered for osteosarcoma or Ewing sarcoma, mainly because of the rarity of the tumor and percentage of successful engraftment and PDX stabilization that varies from 12% [74] to 50% [65] to 100% [77], and a further challenge for the growth of SyS PDX will probably derive from the use of human immune system (HIS) mice [78]. The most relevant SyS PDX collections published in the literature are reported in Table 4. In addition, SyS PDXs currently available from international repositories and companies outsourcing PDX models are reported in Table 5.

### Innovative Drugs Studied in SyS PDX Models

Several studies have investigated the activity of standard and innovative drugs in vivo in SyS PDX models. Stebbing and collaborators [81] and Izumchenko and collaborators [64], by performing retrospective co-clinical studies involving a similar treatment with ifosfamide or trabectedin in SyS patients and in their PDX models, showed highly concordant responses, both positive and negative, for example, in the case of ifosfamide, between the patient and the corresponding PDX.

In the search for new doxorubicin analogues with reduced systemic toxicity and increased antitumor efficacy, the activity of a tetrapeptidic prodrug of doxorubicin (Phosphonoacetyl-L-alanyl-L-leucyl-L-glycyl-L-prolyl-doxorubicin, PhAc-ALGP-doxorubicin or ALGP-doxo) was investigated against the SyS PDX UZLX–STS7, showing significant anti-tumor activity compared to standard doxorubicin treatment that, on the contrary, marginally affected SyS PDX tumor growth [74].

Isfort and collaborators [75] have highlighted that SS18-SSX-dependent YAP/TAZ signaling is overexpressed in a high proportion of SyS where an IGF-II/IGF-IR signaling loop contributes to aberrant YAP/ TAZ activation through dysregulation of the Hippo effectors LATS1 and MOB1. The therapeutic activity of Verteporfin, a YAP-specific inhibitor that can block the interaction between transcriptional coactivator YAP and transcriptional factor TEAD to repress YAP’s function, has been investigated in vivo, in combination with doxorubicin, in preclinical models represented from an SyS cell line (SYO-1)-derived-xenograft and an SyS PDX (UZLX–STS7) [75] with a significant inhibition of tumor growth in both models. Since a simultaneous activation of several different signaling pathways can sustain growth and malignancy of SyS, future therapeutic approaches need to be built on an integrated signaling network investigation and on multiple patient-derived models able to capture the clinical heterogeneity.

A dependency upon the DNA damage response serine/threonine protein kinase ATR (ataxia telangiectasia and Rad3-related), playing an important role in maintaining genome integrity during DNA replication, was detected in SyS. The ATR inhibitor VX970, in combination with cisplatin, was able to inhibit in vitro the growth of several human SyS cell lines, and, in vivo, to significantly decelerate the growth of HS-SY-II cell-derived xenografts. Efficacy of VX970 was shown also against the SA13412 PDX previously indicated as SyS PDX but later found to be a PNET (primitive neuroectodermal tumor) [83].

A preclinical study using tazemetostat, a small-molecule inhibitor of EZH2, the catalytic subunit of PRC2, and an alternative EZH2 inhibitor, EPZ011989, showed a significant slowdown of tumor growth in one out of two cell line-derived xenografts (Fuji responding and HS-SY-II not responding) and in two out of three SyS PDX models (CTG-0771 and CTG-0331 responding, and CTG-1169 not responding), indicating a heterogenous behavior among SyS in vivo models to EZH2 inhibitors [76]. Similarly, phase I/II clinical trials, employing tazemetostat as single agent, displayed variable outcomes in SyS patients. After treatment with tazemetostat, no objective responses were observed in 33 heavily pretreated SyS patients, where tazemetostat induced, only as a best response, stable disease in 33% of patients [8,10,84]. The heterogeneous responses both in preclinical models and in patients highlight the need for additional biomarkers to identify the patients with the highest dependency from a specific pathway and the best molecular probability to respond to targeted treatments, and, again, of multiple models, representative of different tumor signatures, for testing.

Finally, SyS PDXs are being also used for the establishment of new SyS cell lines [40,85,86] that will be reported in the next paragraph.

Since the use of PDX is indicated as mandatory among the Minimum Preclinical Testing Requirements for the Development of Innovative Therapies [56], an increased offer of SyS PDXs is strongly required. This will support more extensive preclinical drug testing, including the set-up of mouse PDX clinical trials able to enhance predictivity of drug efficacy at the clinical level and to pursue more personalized therapies.

## 5. SyS Patient-Derived Cell Lines and Cell-Derived Xenografts (CDX)

For many years, basic biological research and preclinical pharmacology of human sarcomas have been essentially based on cell lines possibly able to grow in immunodeficient mice. Selective pressure for in vitro establishment and long-term culturing can hamper the predictive ability of cell lines, but a valuable advantage resides in the feasibility of biological studies and of high-throughput analyses with comparatively low costs [87]. To recognize the current landscape of SyS cell lines, publications on SyS cell lines and the Cellosaurus cell line database (Expasy—Cellosaurus, https://www.cellosaurus.org/, accessed on 28 November 2022) were considered. In the Cellosaurus database, SyS resulted to be represented by 41 cell lines of human origin, with only a small panel of 4 cell lines (Aska-SS, Yamato-SS, HS-SY-II, and SW982, the last one not having the SS18-SSX translocation) shared in public cell banks (RIKEN BioResource Research Center (https://web.brc.riken.jp/en/, (accessed on 28 November 2022)) and ATCC Human Cells|ATCC (https://www.atcc.org/cell-products/human-cells#t=productTab&numberOfResults=24, accessed on 28 November 2022). Compared to the entries related to human osteosarcoma or Ewing sarcoma, which are around 150 each [87], SyS cell lines are scarce. Of note, no SyS cell line of mouse origin is listed in the Cellosaurus database. However, at least two murine SyS cell lines are reported in the literature: the murine SyS cell line M5SS1, derived from the mouse conditional SyS model Myf5Cre/SS18-SSX2 [37] and used, for example, for SS18-SSX silencing experiments [25], and the murine SyS cell line SSR3A1 (SS18-SSX2), used, for example, in high-throughput drug screening [52].

Some of the SyS cell lines in the Cellosaurus database were so far rarely used in laboratories and did not even carry the annotation of the fusion gene (SYN-1, SYNb-1, SYNb-2, STSAR-198, STSAR-84, SW1045, A1095, HS 192.T, HS197.T, Hs431.T, Hs 701.t, hSS-005R, HSS-84, RIT-3). The Cellosaurus SyS database includes also the extensively studied SW982 cell line, carrying a BRAF V600E mutation but not the SS18-SSX fusion oncogene or high levels of BCL-2. However, a caution note has been added indicating that it is likely to be mis-classified and could represent instead a different tumor. Conversely, not reported in the Cellosaurus database is the human SyS cell line MoJo [39] that can be of interest because of its harboring of, in addition to the SS18-SSX1 translocation, the NRAS Q61R mutation, and displaying high resistance to Pazopanib. Moreover, MoJo cells are able to grow in immunodeficient mice after intramuscular injection.

Interestingly, the human SyS cell line KU-SS-1 was derived from a PDX at the third in vivo passage [85] and, recently, two additional new SyS cell lines, SS-PDX [40] and ICR-SS-1 [86], not yet included in the Cellosaurus database, were independently obtained from the same publicly available SyS PDX J000104314 distributed from The Jackson Laboratory biorepository [86]. Hopefully the establishment of new, clinically relevant SyS cell lines derived from SyS PDX models will increase in the near future and will expand the collection of SyS in vitro/in vivo preclinical models.

Currently, the panel of the most popular SyS cell lines includes seven cell lines: Aska-SS, HS-SY-II, Yamato-SS, MoJo with SS18-SSX1 translocation, and SYO-1, Fuji, and CME-1 with SS18-SSX2 translocation.

The tumorigenic ability in immunodeficient mice, to the best of our knowledge, has been investigated with positive results only for few cell lines. Numbers of injected cells range from 1 to 20 million per mouse and the use of Matrigel can be required. Tumor latency can range from 2 to over 10 weeks and sizeable tumor growth can take over 2–5 months. Table 6 summarizes the main features and tumorigenic ability in immunodeficient mice of the most used and characterized human SyS cell lines reported in the Cellosaurus database or published up to now.

Since genetic data do not exactly predict the response to anticancer agents, in vitro and in vivo drug screening will remain irreplaceable steps. In this perspective, additional SyS cell lines suitable for high-throughput drug testing are strongly needed. Currently, SyS cell lines and, in general, STS are scarcely represented (around 2%) in large-scale drug-screening tests [116] and SyS could only poorly profit from the latest automated technologies. Therefore, an increase in the availability of clinically representative cell lines is strongly advisable.

Recently, drug sensitivity testing comparing SYO-1, HS-SY-II, and the SyS PDX-derived ICR-SS-1 SyS cell line has provided a comprehensive evaluation of the activity in vitro of standard of care drugs and of 58 small molecule inhibitors against SyS cells. In this study, the ICR-SS-1 cell line was found to be significantly more resistant to doxorubicin (IC50 > 600 nM) compared to the other cell lines, SYO-1 and HS-SY-II (IC50 around 10–30 nM). All three cell lines were resistant to Pazopanib (IC50 > 5 µM). Only three compounds, the dual PI3K-mTOR inhibitor NVP-BEZ235, the PLK1 inhibitor BI 2536, and the BET bromodomain inhibitor JQ1, were effective in all three cell lines, indicating the underlying pathways as possible shared vulnerabilities among SyS [86].

A high-throughput drug screen in human and murine SyS cell lines (Aska-SS, Yamato-SS, MoJo, SYO-1, Fuji, and the murine SyS cell line SSR3A1) of over 900 compounds covering 100 different drug classes including epigenetic modifiers, identified some HDAC inhibitors, such as SB939 and Quisinostat (but not Vorinostat, found to be ineffective also in a clinical trial [10]), and proteasomal targeting agents, as the most effective drug categories in SyS in vitro [51]. HDAC2 was supposed to take part in the MDM2 ubiquitination pathway, acting to maintain, in SyS cells, low levels of MULE (Mcl-1 ubiquitin ligase E3), a ubiquitin E3 ligase exhibiting tumor-suppressor function in synovial sarcoma, by targeting the SS18-SSX oncoprotein for ubiquitination and degradation. Of note, MULE is almost undetectable in synovial sarcomas, and its protein levels increase after addition of MG132 proteasome inhibitor [117]. Epigenetic drugs targeting these mechanisms of ubiquitination and degradation will deserve further evaluation in SyS.

Combination of HDAC and CDK4/6 inhibitors was found able to repress the core oncogenic and SS18-SSX programs in four different SyS cell lines in vitro. Moreover, the combined treatment sensitized SyS cells to T cell reactivity and T cell-mediated cytotoxicity in vitro, but further investigations are needed to evaluate their activity in SyS models in vivo [28].

SyS cell lines were used also for investigating the relative proportion of CBAF, GBAF, and PBAF complexes, highlighting that CBAF relative abundance was significantly reduced in at least five SyS cell lines compared to other non-SyS cell lines. Levels of retained CBAF vary among SyS cell lines and can be related to the extent of proteasomal degradation [23]. BRD9, a specific subunit of GBAF complexes, resulted to be essential for the proper assembly of GBAF complexes, and BRD9 degradation, specifically, was able to disrupt this subclass of SS18-SSX-containing complexes. Moreover, only the targeting of BRD9 with degraders, rather than bromodomain inhibitors, was effective in hampering SyS growth [53].

## 6. Perspectives and Conclusions

New targeted therapeutic approaches are an urgent need in SyS, which remains a fatal disease in the cases of relapse and metastatic progression, also because standard chemotherapy displays only a limited activity.

The current landscape of SyS research, exemplified in Figure 1, reveals that SyS investigations can take advantage of both mouse conditional transgenic models and patient-derived models in vitro and in vivo. Integration of data from all these different models has largely improved the understanding of the very complex SS18-SSX oncogenic mechanisms and of the role of the few recurrent additional molecular alterations. However, the advancements in innovative targeted therapies for SyS are still unsatisfactory, and no targeted therapy or immunotherapy options have reached full clinical employment. At variance with tumors that depend on the activation of genes, such as tyrosine kinase receptors, inducing in the cell a state of oncogene addiction, which offers a single actionable target that can be successfully treated with antibodies or inhibitors, in SyS the situation is more complex. In the case of SyS, the fusion oncogene encodes a fusion protein that interact with chromatin remodeling complexes, giving rise to a simple diploid genotype, but a complex dysregulation of gene expression. The multiple activations of several growth factors and signaling pathways requires, therefore, a drug having a broad inhibitory action or multiple targeting of relevant pathways.

In this context, the use of epigenetic drugs determining GBAF degradation such as selective degraders of BRD9 [23,53,54] have produced interesting preclinical results and are at present under evaluation in clinical trials.

Other investigations are focused on immunotherapeutic strategies. The transfer of SPEAR T cells directed against NY-ESO-1 antigen are currently being assessed in SyS therapy [61] and have reached noteworthy results in clinical trials.

Improvements in SyS therapy are being expected in the near future from several directions, and both epigenetic drugs and immunotherapeutic approaches seem to be particularly promising.

The frequent divergence between preclinical efficacy and actual clinical outcomes creates a challenge for improving preclinical modeling.

The growing possibility of investigating new therapeutic approaches for SyS in a wide range of different models, both in vitro and in vivo, will better define the clinically relevant molecular dependencies of SyS. The integration of data from different preclinical models (cell lines, GEMM, PDX) [56] will hopefully increase the clinical predictivity.

In conclusion, even if cross-species challenges, particularly for targeted strategies, will have to be carefully considered, the immunocompetent SyS mouse models and/or SyS PDX could provide valuable tools for exploring additional immunotherapeutic and epigenetic strategies or combinations, and identify molecular markers of sensitivity/resistance to innovative therapies, leading to a better selection of the patients who will benefit from selected targeted therapies, with the aim of conceiving new therapeutic options for SyS and consistently foreseeing the clinical outcome.

## Figures and Tables

**Figure 1 cancers-15-00588-f001:**
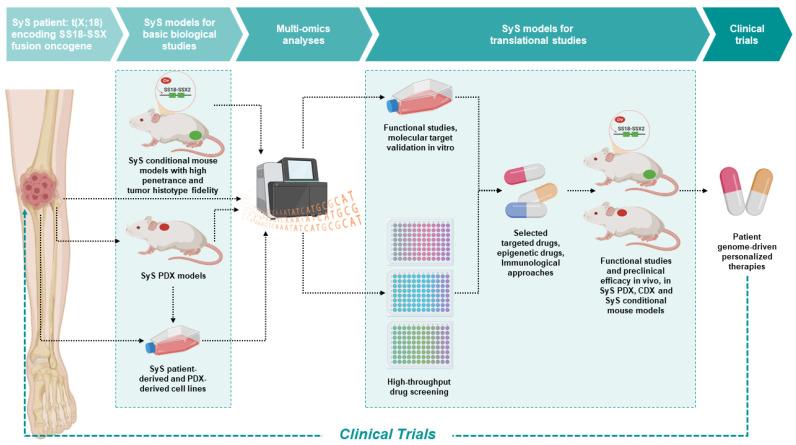
Modeling SyS for the development of targeted, genome-driven, therapeutic approaches. The research on SyS, proceeding from biological and molecular investigations to translational studies, can take advantage of: (I) conditional mouse models, based on Cre technology, carrying a floxed SS18-SSX fusion gene and developing SyS closely resembling human SyS; (II) SyS PDX models; (III) SyS cell lines for in vitro and in vivo studies (cell-derived xenograft, CDX). Molecular and genetic analyses followed by in vitro functional and molecular target validation and high-throughput drug screening can lead to the selection of the most promising drugs to be evaluated for preclinical efficacy in vivo in multiple PDX models (PDX preclinical trials) in immunodeficient mice and/or in SyS conditional mouse models in immunocompetent mice. The final aim is the design of new personalized genome-driven therapeutic options for SyS patients (created with Biorender.com).

**Table 1 cancers-15-00588-t001:** SyS conditional genetically engineered mouse models.

Genotype	Tumor Histology or Main Phenotype	Tumor Incidence	Tumor Latency	Main Features and Notes	References
Rosa26/hSS2 or Rosa26/hSS1	No tumor	None	Not applicable	Carrying a floxed SS18-SSX2 or SS18-SSX1 allele at the Rosa26 locus Viable, fertile, no tumor formation	[37,39]
hSS1 or hSS2 mice +TATCre recombinant protein injection in hind limb	SyS	40–60%	>1 year median 18.8 months	Induction of SyS after Cre protein administration	[39]
Homozygous hSS2 mice +TATCre recombinant protein injection in hind limb	SyS	100%	6 months	Dose dependent increase in SyS onset	[23]
Myf5–Cre/hSS2	SyS	100%	3–5 months	-Multiple tumors (3–5/mouse) of extremities and rib cage -Monphasic >> biphasic -Myogenin-neg, Cytokeratin-pos, Vimentin-pos -Gene expression signature “SS18-SSX model synovial subset” shared by human and murine SyS is present only in human SyS, not in other STS	[37]
*Pax3–Cre*/hSS2 or *Pax7–Cre*/hSS2 or *Hprt–Cre*/*hSS2* or *Hprt–Cre/hSS1* or *Ap2–Cre*/hSS2 or Sox9–Cre/hSS2 or Dermo1–Cre/hSS2 or Flk1–Cre/hSS2 or Tie2–Cre/hSS2 or Nestin1–Cre/hSS2 or Prx1Cre/hSS2	Embryonic lethality			Expression in earlier ectodermic, neural, or mesenchymal precursors, or bone and cartilage precursors, or vascular and hematopoietic precursors, disrupted embryogenesis	[6,37,38,39]
Col1a1Cre/hSS2 or OcCre/hSS2	Perinatal lethality			Bone and skeletal defects No tumors	[6]
*Myf6*–Cre/hSS2	Severe myopathy No tumor			Died within 6 months of age	[37]
Prx1Cre^ERT2^hSS2 +tamoxifen	SyS	100%	9 months	Perimandibular SyS	[6]
OsxCreERT hSS2 +tamoxifen	No tumor			No tumors or skeletal defects up to 1 year	[6]
Rosa26CreER/hSS2	SyS	100%	5–14 months	3 tumors/mouse with a distinct anatomical distribution compared to Myf5–Cre/hSS2 mice	[38]
Myf5–Cre/hSS1 or Rosa26CreER/hSS1	SyS	80–50%	>1 year	Low multiplicity	[39]

**Table 2 cancers-15-00588-t002:** Combination of conditional SyS mouse models and oncogene silencing/expression.

Genotype	Tumor Histology or Main Phenotype	Tumor Incidence	Tumor Latency	Main Features and Notes	References
hSS2/TATCre recombinant protein injection in hind limb+ Bcl2 overexpressing (Bcl-2OE) mice	SyS	80%	Median 9 months	Enhanced synovial sarcomagenesis, increased incidence, and reduced tumor latency compared to hSS2 mice/TATCre	[41]
Myf5–Cre/hSS2 + Bcl2 overexpression	SyS	100%	Earlier onset	Significantly reduced tumor latency	[41]
Myf5–Cre/hSS2, Bcl2fl/fl (Bcl2 deleted)	SyS, monophasic subtype only	80%	5–9 months	Slightly later onset and lower multiplicity compared to Myf5 –Cre/hSS2 mice	[41]
SSM2+/B-CATfl+/+/Myf5–Cre+	Strong inhibition of synovial sarcomagenesis by β-catenin silencing	20%	NA	Strong β-catenin nuclear signal in the few developing tumors, indicating incomplete silencing	[43]
Rosa26hSS2/wt; Ctnnb1ex3^fl/wt^ +AdCre injection	Increased synovial sarcomagenesis; poorly differentiated SyS subtype	90%	3 months	Beta-catenin stabilization, increased SyS sarcomagenesis No metastasis	[44]
*Myf5–Cre*; *Ctnnb1ex3fl*	Embryonic lethality				[44]
*Prx1CreERT2 hSS2 Ctnnb1ex3fl* *+tamoxifen*	SyS in the forelimbs	100%	3 months		[6]
*OsxCreERT* *Ctnnb1ex3fl +tamoxifen*	SyS	100%	NA	Osteopetrosis	[6]
hSS2 Myf5–Cre/Fgfr1,2,3^fl/fl^ (SMF1,2,3.HO) and each single Fgfr knockout (FGFR silencing)	Significantly reduced SyS incidence and multiplicity	10–35%	Observed at 10 weeks	FGFR homozygous (HO) silencing. Implication of mechanisms related to ETV4 and ETV5 through DUX4 embryonic pathway	[47]

NA, not available.

**Table 3 cancers-15-00588-t003:** Combination of conditional SyS mouse models and tumor suppressor gene silencing.

Genotype	Tumor Histology or Main Phenotype	Tumor Incidence	Tumor Latency	Main Features and Notes	References
hSS1 or hSS2/*PTEN^fll^*^fl^ +TATCre injection in the hind limb (PTEN silencing)	Increased SyS incidence and acquisition of metastatic potential to the lung, metastatic SyS	100%	<1 year (8–10 months)	-TATCre induction of homozygous silencing of PTEN alone induced no tumors -40% incidence of lung macrometastses -70% incidence of lung micrometastases ->90% incidence of lung disseminated tumor cells	[48,51]
hSS2/SMARCB1^fl/fl^ +TATCre injection (SMARCB1 silencing)	Increased sarcomagenesis, but development of epithelioid sarcoma or mesenchymal rhabdoid tumor, not SyS	100%	3 months	BAF-family complexes perturbation	[23]

**Table 4 cancers-15-00588-t004:** SyS PDX models in most relevant published collections.

Number of Established * SyS PDX Models	Mouse Strain	Rate of Engraftment (PDX/Implanted Tumors)	PDX Model ID and Annotations	References
6	athymic nu/nu mice	100% (6/6)	Histological evaluation	[77]
2	athymic NMRI-nu/nu mice	NA	PDX ID: S.Lt, S.To Histological evaluation	[79]
1	athymic nu/nu mice	100% (1/1)	0.91 Pearson correlation coefficient between the originating patient tumor and the PDX, based on Affymetrix gene expression data	[80]
1	athymic nu/nu mice	50% (1/2)	PDX ID: 1152, SS18-SSX fusion High drug sensitivity to Ifosfamide and trabectedin, intermediate sensitivity to gemcitabine and pazopanib	[81]
2	athymic nu/nu mice	100% (2/2)	PDX ID: CTG-0771, SS18-SSX2 fusion PDX ID: CTG-1169, SS18-SSX1 fusion	[64]
1	NSG, athymic nu/nu mice	50% (1/2)	Pediatric SyS, time to passage 8 months	[65]
1	athymic NMRI-nu/nu mice	12% (1/8)	PDX ID: UZLX-STS7 SS18-SSX1 fusion, Poorly differentiated subtype, over 48 passages, growth rate 1 month	[74]

* At least two in vivo passages; NA, not available.

**Table 5 cancers-15-00588-t005:** SyS PDX models in international repositories.

International Repositories and Internet Links ^§^	Number of SyS PDX Models	PDX Model ID, Annotations	References
Champions Oncology Model Cohorts (championsoncology.com) (https://www.championsoncology.com/resource-library/model-cohorts, (accessed on 28 November 2022))	3	CTG-0771, SS18-SSX2 fusion CTG-1169, SS18-SSX1 fusion CTG-0331, SS18-SSX2 fusion	[64,76]
Xenosarc Platform (Leuven, Belgium) XenoSarc platform—Laboratory of Experimental Oncology (kuleuven.be, (accessed on 28 November 2022)) (https://gbiomed.kuleuven.be/english/research/50488876/50488902/xenosarc, (accessed on 28 November 2022))	1	UZLX –STS7, SS18-SSX1 fusion	[74,75]
Crown Bioscience Patient-Derived Xenograft—PDX Models|Crown Bioscience (https://www.crownbio.com/model-systems/in-vivo/pdx-models, (accessed on 28 November 2022))	3	SA10159 SA10162 SA10175 [SA13412 previously diagnosed as SyS but later diagnosed as primitive neuroectodermal tumor (PNET)]	[82,83]
NCI Patient-Derived Models Repository (PDMR) PDCM Finder—Search (cancermodels.org, (accessed on 28 November 2022)) (https://www.cancermodels.org/data/search?q=Synovial%20Sarcoma, (accessed on 28 November 2022)) Including Childhood Solid Tumor Network www.stjude.org/CSTN/, (accessed on 28 November 2022) (http://www.stjude.org/CSTN, (accessed on 28 November 2022))	12 from 4 sources (JAX, SJCRH, WUSTL, PDMR)	PDMR/119177-322-R1 PDMR/197587-005-T PDMR/761936-265-R PDMR/571681-099-R PDMR/957923-259-R WUSTL/WUSTL SHIM9 WUSTL/WUSTL SHIM11 WUSTL/WUSTL SHIM12 SJCRH/SJSS049190_X1 SJCRH/SJSS063828_X1 JAX/J000104314 (SS18-SSX1 fusion, monophasic subtype) JAX/TM01634 (SS18-SSX1 fusion)	[40,65]

^§^ Accessed on 28 November 2022; (Abbreviations: JAX, The Jackson Laboratory; SJCRH, St. Jude Children’s Research Hospital; WUSTL, Washington University in St. Louis; PDMR, NCI National Cancer Institute Patient-Derived Models Repository).

**Table 6 cancers-15-00588-t006:** Most used human SyS cell lines SS18-SSX translocated and CDX mouse models.

Cell Line	Fusion Gene	Histology	Tumorigenic Ability in Mice and Annotations	References and Banks
A2243	SS18-SSX2	Biphasic	NA	[18]
ASKA-SS	SS18-SSX1	Biphasic	In BALB/c nu/nu, 1000–1 × 10^7^ cells sc, tumor incidence was 100% within 5 months	[88] Cell Engineering Division-CELL BANK-(RIKEN BRC) (https://cell.brc.riken.jp/en/, (accessed on 28 November 2022))
CME-1	SS18-SSX2	Monophasic	In SCID mice, 20 × 10^6^ cells im	[89,90,91,92]
Fuji	SS18-SSX2	Monophasic	In Balb/C-nu, 1 × 10^7^ cells sc 50% Matrigel 200 mm^3^ at 25 days after cell injection	[18,93]
FU-SY-1	SS18-SSX1	Monophasic	Not tumorigenic	[94]
GUSS-1	SS18-SSX1	Biphasic	NA	[95]
GUSS-2	SS18-SSX1	Monophasic	NA	[95]
GUSS-3	SS18-SSX1	Biphasic	NA	[95]
GUSS-3b	SS18-SSX1	Biphasic	NA, deriving from the same patient of GUSS-3 after neoadjuvant chemotherapy and radiation	[95]
HS-SY-II	SS18-SSX1	Monophasic	In Balb/C-nu, 1 × 10^7^ cells sc 50% Matrigel 170 mm^3^ at 35 days after cell injection	[25,96,97] Cell Engineering Division-CELL BANK-(RIKEN BRC) (https://cell.brc.riken.jp/en/, (accessed on 28 November 2022))
HS-SY-3	SS18-SSX1 truncated	Monophasic	Not tumorigenic in nude mice	[98]
ICR-SS-1	SS18-SSX1	Monophasic	NA	[86] Not included in Cellosaurus
KU-SS-1	SS18-SSX2	Monophasic	In SICD mice, 8 × 10^7^ cells sc, tumor latency 16 weeks. Derived from a PDX at the third in vivo passage	[85]
MoJo	SS18-SSX1	Monophasic	In SCID mice, 20 × 10^6^ cells im, tumor growth within 60 days from cell injection Resistant to pazopanib both in vitro and in vivo Harbor the NRAS Q61R mutation	[39,92,99]
NCC-SS1-C1	SS18-SSX1	Poorly differentiated	NA	[100]
NCC-SS2-C1	SS18-SSX2	Poorly differentiated	NA	[101]
NCC-SS3-C1	SS18-SSX1	Monophasic	NA	[102]
NCC-SS4-C1	SS18-SSX1	Monophasic	No in Balb/C-nu, 1 × 10^6^ cells sc 50% Matrigel	[103]
NCC-SS5-C1	SS18-SSX1	Poorly differentiated	NA	[104]
PDSS-26	SS18-SSX1	Poorly differentiated, small cell variant	NA	[105]
SCS214	SS18-SSX2	NA	NA	Cellosaurus SCS214 (CVCL_WU91)
SN-SY-1	SS18-SSX1	Monophasic	In Balb/C-nu, 1.3 × 10^7^ cells sc 33% positive after 23 weeks from cell injection	[106]
SS.PDX	SS18-SSX1	Monophasic	NA	[40] Not included in Cellosaurus
SS255	SS18-SSX2	Monophasic	NA	[18,107,108]
SYO-1	SS18-SSX2	Biphasic	Yes, 5 × 10^6^ sc In NSG or SCID mice or 10^5^ cells/mouse inBALB/c nu/nu SYO-1 cells harbor mutation in CTNNB1 (G34L) with nuclear accumulation of Beta-catenin	[24,25,92,109,110,111]
YaFuSS	SS18-SSX1	Monophasic	NA	[25,46,112]
Yamato-SS	SS18-SSX1	Biphasic	In BALB/c nu/nu 1000–1 × 10^5^ cells sc, tumor incidence was 100% within 5 months; 1 × 10^7^ cells sc tumor latency 2 weeks	[88] Cell Engineering Division-CELL BANK-(RIKEN BRC) (https://cell.brc.riken.jp/en/, (accessed on 28 November 2022))
1273/99	SS18-SSX2	NA	NA	[90,113,114]
716 SS MNV	SS18-SSX	NA	NA	[115]

NA, not available; sc, subcutis; im intramuscular; adapted from Cellosaurus database https://www.cellosaurus.org/; Cellosaurus query synovial sarcoma human, accessed on 22 November 2022.

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
