# Peer review of "Synovial Sarcoma Preclinical Modeling: Integrating Transgenic Mouse Models and Patient-Derived Models for Translational Research"

_cancers, 2023, doi:10.3390/cancers15030588_

Round 1

Reviewer 1 Report

Authors provide a broad review of the pre-clinical models available for synovial sarcoma research, the improvements in the knowledge of the biology and of the sensitivity of the tumor to treatments achieved through their use, and a perspective on the future advancements in this research field thanks to the growing number of models. The review is interesting and well written.

Some minor comments:

Please check uniformity of the use of abbreviations along the text eg. Sys in line 71 should read SyS.

Line 75. Authors may wish to add a range of response of SyS to chemotherapy.

Table 1. Rosa26/hSS2 or Rosa26/hSS1. Since no tumors were developed, tumor latency should report not applicable in place of none.

Lines 253-254 and lines 258-261. Authors may consider rephrasing.

Lines 255-258. Please confirm correct spelling of BCLxL.

Lines 272-257, please rephrase to clarify the take home message and the difference between the two chunks of the sentence. Authors state that “β-catenin stabilizing mutations and nuclear β-catenin staining have been reported in 30–70% of SyS” then “APC-inactivating and CTNNB1-stabilizing mutations are present in some human SyS,…”

Lines 293-294. How this information fits with lines 272-273 stating “β-catenin stabilizing mutations and nuclear β-catenin staining have been reported in 30–70% of SyS”?

Line 303. “dependencies” should read “dependency”.

Line 332. FGFR already abbreviated.

Lines 428-430, please check fonts.

Line 678. “have produce” should read “have produced”.

Line 681. SPEAR already abbreviated before.

Author Response

Reviewer 1:

  • Please check uniformity of the use of abbreviations along the text eg. Sys in line 71 should read SyS.

Abbreviations have been checked and changed accordingly throughout the manuscript (lines 72, 118,145, 234, 556)

  • Line 75. Authors may wish to add a range of response of SyS to chemotherapy.

A sentence has been added. See line 77

  • Table 1. Rosa26/hSS2 or Rosa26/hSS1. Since no tumors were developed, tumor latency should report not applicable in place of none.

Table 1 has been changed accordingly

  • Lines 253-254 and lines 258-261. Authors may consider rephrasing.

Some sentences have been changed. See lines 261-263 and 266-270

  • Lines 255-258. Please confirm correct spelling of BCLxL.

BCL-XL was used. See lines 247 and 264-270

  • Lines 272-257, please rephrase to clarify the take home message and the difference between the two chunks of the sentence. Authors state that “β-catenin stabilizing mutations and nuclear β-catenin staining have been reported in 30–70% of SyS” then “APC-inactivating and CTNNB1-stabilizing mutations are present in some human SyS,…”

The sentence has been corrected and clarified. See lines 280-284

  • Lines 293-294. How this information fits with lines 272-273 stating “β-catenin stabilizing mutations and nuclear β-catenin staining have been reported in 30–70% of SyS”?

We corrected the previous point that now fit with this sentence, lines 303-305, that was left unmodified

  • Line 303. “dependencies” should read “dependency”.

The word was corrected. See line 312

  • Line 332. FGFR already abbreviated.

The repeated abbreviations were corrected. See lines 340 and 342

  • Lines 428-430, please check fonts.

Fonts were changed. See lines 439-440

  • Line 678. “have produce” should read “have produced”.

The word was corrected. See line 689

  • Line 681. SPEAR already abbreviated before.

The repeated abbreviations were corrected. See line 692

Some minor orthographic errors were corrected. See lines 106, 116, 239, 665, 708

In addition we had to change the affiliation of Pier-Luigi Lollini and Francesca Ruzzi because the University of Bologna at the beginning of the new year changed the name of their affiliation Department. See Authors’ affiliation lines 8-9 and 12-13.

Reviewer 2 Report

Dear Editor,

Thanks for the opportunity to review the manuscript titled, “ Synovial sarcoma preclinical modeling: integrating transgenic mouse models and patient-derived models for translational research"

Recent advances in medical research have shown that translational research is key to the understanding of diseases and treatments. Synovial sarcoma is a devastating form of cancer both in children and adults with a very high mortality rate. The manuscript resumes the literature and makes the reader to understand the importance of the experimental in vivo models for a rare tumor such as synovial sarcoma.

The authors made a thoughtful review that examined systematically all the recent literature in this field, dividing the chapter based on the type of models (transgenic mouse, patient-derived xenograft, patient-derived cell lines and cell-derived xenografts).

The chapter dealing with the transgenic mouse (chapter 3) also revealed molecular patterns that are involved in the specific models of the related studies, in some cases, mentioning briskly if these molecular alterations are also present in human synovial sarcomas.

A better correlation of the tumor developed in the transgenic mouth models with synovial sarcomas of patients would be more desirable. However, the authors cannot be blamed, since most of the mentioned studies were focused on their model without actually looking, in most of cases, for histopathological and clinical reproducibility.

I would suggest to add in the introduction some of the limits of this type of research caused, as the authors pointed out, by the absence of the knowledge of a cell origin in synovial sarcomas as in other sarcomas.

There are also some minor issues that the authors need to address before the manuscript can be considered for publication, mostly with chapter 6. Some sentences are confusing and consequently, make it difficult to understand what the authors mean to say (Example line 672). This paragraph should be revised and some of the phrases should be improved to make it more understandable to the readers. I would recommend also to put the perspective and conclusions in one chapter and make some adjustments.

The review paper paves the way for future researchers to find more effective treatments for synovial sarcomas and research on more in vivo animal models of the disease. Overall I found it to be relevant, and I would consider this paper, after some minor revision, to be published.

Best regards,

Author Response

Reviewer 2:

  • A better correlation of the tumor developed in the transgenic mouth models with synovial sarcomas of patients would be more desirable. However, the authors cannot be blamed, since most of the mentioned studies were focused on their model without actually looking, in most of cases, for histopathological and clinical reproducibility.

I would suggest to add in the introduction some of the limits of this type of research caused, as the authors pointed out, by the absence of the knowledge of a cell origin in synovial sarcomas as in other sarcomas.

A sentence has been added regarding limitations of preclinical SyS modeling. See lines 138-144

  • There are also some minor issues that the authors need to address before the manuscript can be considered for publication, mostly with chapter 6. Some sentences are confusing and consequently, make it difficult to understand what the authors mean to say (Example line 672). This paragraph should be revised and some of the phrases should be improved to make it more understandable to the readers. I would recommend also to put the perspective and conclusions in one chapter and make some adjustments.

A single chapter Perspectives and Conclusions was created, and sentences were clarified. See lines 668, 682-687, 709, 714-716, 718

Some minor orthographic errors were corrected. See lines 106, 116, 239, 665, 708

In addition we had to change the affiliation of Pier-Luigi Lollini and Francesca Ruzzi because the University of Bologna at the beginning of the new year changed the name of their affiliation Department. See Authors’ affiliation lines 8-9 and 12-13.
